# Real-World Evaluation of the Tolerability to Onabotulinum Toxin A: The RETO Study

**DOI:** 10.3390/toxins14120850

**Published:** 2022-12-03

**Authors:** David García-Azorín, Blanca Martínez, María Gutiérrez, Marina Ruiz-Piñero, Ana Echavarría, Álvaro Sierra, Ángel L. Guerrero

**Affiliations:** 1Headache Unit, Neurology Department, Hospital Clínico Universitario de Valladolid, 47005 Valladolid, Spain; 2Department of Medicine, University of Valladolid, 47002 Valladolid, Spain

**Keywords:** onabotA, chronic migraine, adverse effects

## Abstract

Onabotulinumtoxin A (onabotA) has shown efficacy in chronic migraine (CM), with good tolerability and a low rate of adverse effects, most of them not severe. The aim of this study is to evaluate tolerability and adverse effects of onabotA in clinical practice and to analyze if there is a relationship between tolerability to treatment administration, adverse effects’ (AEs) occurrence and clinical response. We included patients with CM that received treatment with onabotA for the first time. Tolerability to treatment was evaluated by a 0–10 numeric rating scale (0: worst possible, 10: optimal tolerability). We assessed the presence of AEs by using a standardized questionnaire. Treatment response was based on the 50 and 75% responder rate between weeks 20 and 24, compared with the baseline, according to headache diaries. We analyzed whether the tolerability was associated with a higher frequency of AEs or a higher probability of clinical response. We included 105 patients, 87.7% female, with an age of 43.9 ± 10.7 years. Mean tolerability was 7.8/10 and 7.2/10 in the first and second onabotA administration, respectively. AEs were reported by (first-second) 71.4–68.6% patients. The percentage of patients with a 50% response was 56.3%. There was no association between tolerability and AEs’ occurrence or clinical response.

## 1. Introduction

Migraine is the main cause of years lived with disability between 15 and 49 years [1,2] and global prevalence of Chronic Migraine (CM) is 1–2% [3]. The impact in patients’ lives reached the maximum in those with chronic migraine, due to the higher frequency of comorbidities, monetary costs and concomitant medication-overuse headache [4,5,6,7,8]. CM has negative effects on global functioning of the patients as well as a reduced health-related quality of life [3]. To date, topiramate [9,10,11], onabotulinumtoxinA (onabotA) [12,13,14,15,16] and monoclonal antibodies against Calcitonine Gene-related Peptide (CGRP-mAbs) [17,18,19,20] have proven efficacy and have tolerability as a prophylactic treatment for CM in randomized clinical trials (RCT). However, oral treatments are often discontinued due to a lack of efficacy or poor tolerability. Onabot A has shown cost-effectiveness when treating the adult population with CM [21]. 

The clinical results are not fully comparable, because the designs of the RCT were not identical [9,10,11,12,13,14,15,16,17,18,19,20]. Probably, the most remarkable differences between them are the way of administration and the tolerability profile. OnabotA could decrease peripheral and, indirectly, central sensitization [22,23]. Its administration is based on Phase III REsearch Evaluating Migraine Prophylaxis Therapy 1 (PREEMPT) Paradigm [12], which includes 31 injections in the cranial and cervical region. Compared with oral prophylactic treatments or mAbs, patients might perceive onabotA administration as more unpleasant. However, studies specifically analyzing the tolerability to onabotA administration are scarce. 

The aim of this study was to systematically evaluate the tolerability to onabotA administration and the frequency and type of adverse events (AE). We analyzed whether the tolerability was associated with the presence of AEs or with the clinical response. 

## 2. Results

During the study period, 106 patients were included. One patient was excluded because they received an anesthetic blockade prior to the first onabotA, for a final sample of 105 patients. The mean age was 43.9 ± 10.7 years and 92 (87.6%) patients were female. Median age of migraine onset was 17 years (IQR: 13.0 - 24.0) and the median of months with chronic migraine was 18.0 (IQR: 10.5 - 41.0) months. 

Frequency of headache days at baseline was 23.9 ± 5.3 days per month and for migraine headache days it was 12.2 ± 6.7 days per month. Frequency of acute medication uptake at baseline and triptan use was 18.1 ± 7.7 and 6.9 ± 7.1 days per month, respectively. Medication-overuse headache criteria were met by 82 (78.1%) patients at baseline.

At the moment of onabotA administration, 75 (71.4%) patients in the first session and 69 (65.7%) patients in the second session were experiencing headache, with a mean intensity of 4.8/10 ± 2.1 and 5.6/10 ± 2.3, respectively.

The onabotA administration protocol was modified in 42 patients (40.0%) in the second session. We increased the dose according to the follow-the-pain strategy in 33 patients, and we added a greater occipital nerve anesthetic blockade in 8 cases after tolerability assessment. Cervical infiltration points were suppressed in five cases, and the corrugator infiltration points were elevated in four patients. 

### 2.1. Tolerability

The mean tolerability was 7.8/10 in the first session and 7.2/10 in the second. The tolerability mean worsened to 0.6 ± 2.1 points at the second session compared to the first (paired Student *t* test, *p* = 0.002). The mean tolerability ratings are summarized in Figure 1. One patient experienced syncope during the first administration (0.9%) and another patient had a presyncope in the second session (0.9%). 

Tolerability to the first administration showed a moderate correlation with tolerability to the second administration (*p* = 0.47, *p* < 0.001) and a weak correlation with age (R = 0.26, *p* = 0.006). In the regression analysis, age was the only variable that was associated with better tolerability. Tolerability was not associated with headache intensity at the moment of the infiltration (*p* = −0.86, *p* = 0.38) or with frequency of headache at baseline (*p* = 0.11, *p* = 0.23). Table 1 summarizes the analysis of variables associated with tolerability. 

### 2.2. Adverse Event Occurrence

AEs were described by 75 (71.4%) patients after the first session and by 72 (68.6%) patients after the second. All AEs were graded one (mild but not bothersome) or two (bothersome but not dangerous). There was no statistical difference between the frequency of AEs of each session (Wilcoxon test, *p* = 0.48). Frequency of headache was the only AE frequency which showed a statistical difference between both sessions (32.3% vs. 45.7%, McNemar test, *p* = 0.015). (Figure 2). The longer duration of CM and the higher number of headache days per month at baseline were associated with the presence of AEs (Table 2). 

### 2.3. OnabotA Response 

Figure 3 shows frequency of headache days, intense headache days, symptomatic medication days and triptan days per month at baseline, after 8–12 weeks and after 20–24 weeks. A 50% response rate was observed in 56 (53.35) patients between weeks 8 and 12 and 52 (49.5%) between weeks 20 and 24. A 75% response was observed in 37 (35.2%) patients between weeks 8 and 12 and 30 (28.6%) between weeks 20 and 24. 

Patients with 50% response between weeks 8 and 12 had a lower mean intensity of headache at the moment of infiltration (2.9 vs. 4.1, *p* = 0.02). Mean tolerability in responders was 7.2/10 compared with non-responders at 7.9/10, but these differences did not reach statistical signification (*p* = 0.052). Patients with 50% response between weeks 20 and 24 had lower headache intensity at the moment of onabotA administration (2.8 vs. 4.5, *p* = 0.009). Tolerability did not differ between responders and non-responders (6.8 vs. 6.9, *p* = 0.69). 

## 3. Discussion

OnabotA is an effective, safe and well-tolerated prophylactic treatment for migraines [24]. It is also effective and safe in elderly patients [25] and improves quality of life in CM patients [26].

The exact mechanism by which OnabotA has an antinociceptive effect has not been fully elucidated. Onabot A seems to block the peripheral release of neuropeptides involved in neurogenic inflammation as Substance P and CGRP. Additionally, it also blocks the translocation of membrane receptors to the surface of sensory neurons, such as the vanilloid transient receptor potential channel (TRPV1) [21].

However, many patients with CM are reluctant and have some fear about preventive treatments, and this may be caused by its way of administration, based on pericranial and cervical injections [27]. Most trials and real-world studies describe tolerability within the secondary endpoints, not always in a systematic way, and evaluate the immediate tolerability to the treatment administration [28]. In the present study, we aimed to focus on the tolerability and AEs’ occurrence. The main findings were that onabotA administration was well tolerated and AEs were frequent but mostly local and mild. 

We assessed tolerability in the two first sessions in onabotA naïve patients by immediately asking them after the injection. Participants judged the tolerability as good, similar to previous studies’ results [12,28,29], with a mean tolerability score of 7.8/10 in the first session and 7.2/10 in the second. We completed the evaluation by asking the clinician and nurse who participated in the infiltration, and the relative. As expected, their score was more optimistic than the patients’ evaluation, but the results were quite similar, and the scores remained stable within the two sessions. Interestingly, the second session was worse tolerated than the first one, although patients presented fewer AEs, except for headache. This could be related to the dose increase during the second session, but further investigation should be addressed.

Several factors have been associated with a better clinical response of OnabotA [30,31,32,33], but there is less information regarding tolerability not influenced, according to previous studies, by circadian time of administration [34] or needle length [35]. In our study, the only factor associated with a better tolerability was age. However, as this result is a consequence of an exploratory analysis, it may be interpreted cautiously and further large studies should specifically evaluate the tolerability predictors. 

Another relevant contribution of the present study was the systematical and prospective evaluation of adverse effects. The pooled analysis of CM onabotA pivotal studies [16] showed an AE rate of 71,4%, mostly mild events, the most frequent being neck pain (13.8%), followed by muscular weakness (9.2%), headache (8%), facial paresis (7.9%) and musculoskeletal stiffness (7.2%). One of the largest real-world studies is a prospective observational European study [29], which showed a rate of ≥1 treatment-related AEs of 25.1%. The most commonly reported treatment-related AEs were neck pain (4.4%), eyelid ptosis (4.1%), muscular weakness (2.9%), headache (2.5%), musculoskeletal stiffness (2%) and migraine (2.9%). Another large study, the COMPEL study [34], showed a treatment-related AE rate of 18.3%. The most frequent AEs in this study were neck pain (4.1%), followed by eyelid ptosis (2.5%), musculoskeletal stiffness (2.4%), pain in injection side (2%), headache (1.7%), muscular weakness (1.4%), facial palsy (1.3%), migraine (1%) and skin tightness (1%). In our study, we found a higher AE rate when compared with pivotal and real-life studies. After the first session, 75 patients (71.4%) described some AEs, and after the second administration, 72 patients (68.6%) reported some. We did not find any severe AEs. The most frequent AE in our study was headache (32.4% in the first session, 45.7% in the second), followed by aesthetic effect (25.7–19%), cervical stiffness (10.5–7.6%), muscular pain (8.6–2.9%) and eyelid ptosis (3.8–1.9%). No patients discontinued the treatment during the study because of AEs. This higher rate of AEs compared to other studies might be mainly in relation to post-procedure headache (32.4–45.7% compared with a rate around 2% in other studies) and the systematic evaluation of AEs 14 ± 4 days after infiltration, instead of at 12 weeks as in most studies [16,36], which could have caused some degree of recall bias.

The tolerability profile of onabotA differs from its main alternatives, topiramate and CGRP-mAbs, being one of the main differences between those therapies. The reported frequency of AEs in clinical trials was 75–82% for topiramate [9,10,11] and 57.9–62.1% for GCRP-mAbs [33], leading to treatment discontinuation in the pivotal RCT in 10.8% [9,10,11] and 1.4% [17,18,19,20], respectively. Topiramate might cause paresthesia (53–28.8%), fatigue (11.9–6%), nausea (9.4–9%), memory and concentration disturbances (9.4–6%) and anorexia (6–5%) [9,10,11]. CGRP-mAbs might cause injection site pain, fatigue, upper respiratory infections and constipation [37].

There are some recognized predictors to OnabotA response [33]. Nevertheless, the relationship between tolerability and clinical response or adverse effects has not been fully studied. We hypothesized that the better tolerability could be related with a higher probability of clinical response or a lower probability of adverse effects. However, we could not find any solid association. This could be explained by the potential lack of power of the study or the absence of relevant association. Indeed, we did observe a worse tolerability in responders when compared with non-responders (7.2 vs. 7.9), albeit these differences did not reach statistical signification. Further studies should be conducted to address the possible association between tolerability and clinical response. 

The present study has some limitations. Firstly, it is a single-center study, with a modest sample size compared to pivotal and real-world studies, thus it would be interesting to include other hospitals from different areas for further studies. Secondly, the proportion of patients with medication-overuse headache was relatively high, though in line with other real-world series [27]. Some of the study endpoints were exploratory and, as a result, the study could be underpowered to detect differences, even whenever present. In our opinion it is advisable to carry out multicentric studies with larger samples and with an evaluation of efficacy depending on whether the procedure was carried out in a patient with or without headache. 

## 4. Conclusions

OnabotA administration was well tolerated by patients during the first two sessions. Adverse effects were frequent, but most of them were local and mild. The most frequently reported AEs were headache and neck stiffness. We did not observe a correlation between tolerability to onabotA administration and the clinical response or the presence of AEs. 

## 5. Materials and Methods

### 5.1. Study Design and Setting

We performed an observational analytic study with a prospective cohort design. The study protocol was reviewed and approved by the local Ethics Review Board (code PI: 17-831). The database is available for other researchers upon reasonable request. This study is reported in accordance with the STrengthening the Reporting of OBservational studies in Epidemiology (STROBE) statement [38].

### 5.2. Eligibility Criteria

The inclusion criteria were: (1) definite chronic migraine diagnosis (headache occurring on 15 or more days/month for more than three months, which on at least eight days/month has the features of migraine headache) according to the International Classification of Headache Disorders (ICHD) [39]; (2) with prior failure to at least two oral prophylactics according to the Spanish Headache Study Group guidelines [40]; (3) first time treated with onabotA; (4) older than 18 years old; (5) agreed to participate and signed informed consent form.

Patients were excluded if they had any contraindications for onabotA, had previously received onabotA for migraine prophylaxis or any other indication, were unable to be followed up or to report their clinical situation. 

### 5.3. Sources and Methods of Selection

Eligibility was systematically evaluated in all consecutive patients that were evaluated in the headache outpatient clinic of Hospital Clínico Universitario, Valladolid, Spain, a third-level hospital with a reference population of 261,000 inhabitants. Patients kept a headache diary starting from the preceding month of the study enrolment and until week 24 after. 

### 5.4. Study Endpoints

The primary endpoint of the study was to determine the patients’ subjective tolerability to the first administration of onabotA. The secondary endpoints included: (1) the evaluation of the subjective tolerability to onabotA administration on the second session, (2) the comparison between the first and second infiltrations, (3) the description of the frequency and type of adverse events. As exploratory objectives, we analyzed (1) whether there was a correlation between tolerability and AEs’ occurrence, (2) whether there was a correlation between tolerability and clinical response, defined by 50% response rate, (3) which demographical or clinical factors predicted a better tolerability. 

### 5.5. Variables

A series of demographical and clinical data were collected by the time of patients’ enrolment through an in-person interview, conducted by a neurologist focused on headache disorders. The evaluated variables included age, sex, age of migraine onset, duration of chronic migraine in months, number of headache days per month, number of intense headache days per month (defined as days with intensity of headache over 7/10, in a numeric rating scale (NRS), number of acute medication days per month, number of triptan use days per month, presence of concomitant medication-overuse headache (according to the International Classification of Headache Disorders (ICHD) criteria [22,23]) and type and number of prior prophylactic drugs. From the prior prophylactic drugs, we analyzed also whether there was a lack of efficacy, when used for a sufficient time and at an adequate dose, or a lack of tolerability. If, at inclusion, the patient was receiving an oral preventive with good tolerance and partial efficacy, it was maintained during the study.

### 5.6. Intervention

OnabotA was injected by two neurologists with experience with headache disorders and more than two years of experience in onabotA injection. The infiltration was conducted and distributed in 31 craniocervical points according to the PREEMPT paradigm [12]. During the first session, patients received a standard dose, 155 U of onabotA. During the second session performed after 12 weeks, the standard dose was increased up to 195 U if the clinical response was inadequate and the neurologist deemed it appropriate. 

In each session, patients were asked to answer whether they presented headache at the moment of the infiltration, and, whenever present, they rated the intensity according to a 0–10 NRS (0: no pain, 10: worst possible pain). Immediately after the onabotA administration, patients were asked to rate how well they tolerated the procedure by using a 0–10 NRS (0: worst possible tolerability; 10: best possible tolerability). In addition to patients’ opinion, three additional evaluators judged patient’s tolerability, including the clinician who performed the injection, the nurse who assisted the procedure and, whenever present, the patient’s relative. 

Patients were contacted by phone 14 ± 4 days after each procedure. They were inquired about AEs, first inviting them to freely report any possible AEs and later being systematically inquired about the presence of the most frequent AEs [12,13,14,15,16], according to a pre-defined questionnaire, including a new headache episode or a worsening of the already presented headache during the day or the day after the onabotA administration; eyelid ptosis; aesthetic effects, such as a raised eyebrow or a feeling of not being able to move the forehead; cervical stiffness; and muscular pain.

During the study period, patients completed a headache diary, which included headache days, intense headache days, acute medication days and triptan use days. The response to onabotA was defined by the 50% response rate as decrease of at least 50% in the number of headache days per month between weeks 20 and 24, compared with the month prior to the onabotA administration [41]. We also analyzed the 50% response rate between weeks 8 and 12 and the 75% response rate between weeks 8 and 12 and 20 and 24 as secondary variables. Study protocol is summarized in Figure 4.

### 5.7. Statistical Analysis

Data were summarized using descriptive statistics. Discrete variables were expressed as the number of cases and their percentage. Continuous quantitative variables were described as mean and standard deviation or median and inter-quartile ranges, depending on the normal or non-normal distribution. Comparison between categorical variables was performed using the Chi-square test or Fisher’s exact test, adjusting the *p*-value by the Bonferroni method for multiple comparisons correction. Student’s T-test was used to compare categorical and continuous variables and a paired-T test was used to compare these variables between the first and second administration. Correlation between continuous quantitative variables was assessed with Pearson’s test. We created a regression analysis to evaluate which variables were associated with tolerability. After the adequate adjustment, the level of significance was defined as a *p*-value <0.05. Sample size was not estimated. Statistical analysis was carried out using SPSS Statistics version 26 (IBM Corp. Armonk, NY, USA). 

## Figures and Tables

**Figure 1 toxins-14-00850-f001:**
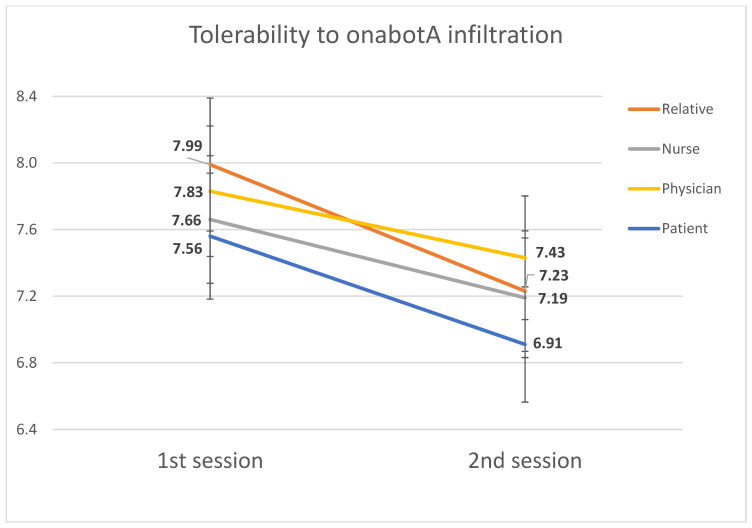
Mean tolerability numerical rating score according to patients, relatives, nurses and clinicians after the first and second onabotA administration.

**Figure 2 toxins-14-00850-f002:**
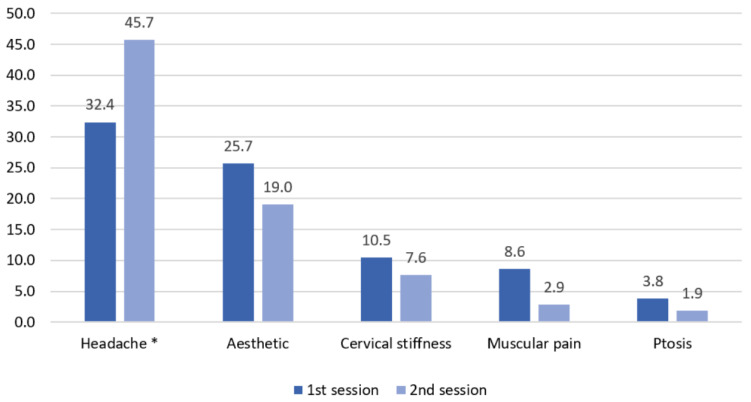
Treatment-related adverse event frequency. Frequency of different adverse effects after each session. * no-migrainous headache.

**Figure 3 toxins-14-00850-f003:**
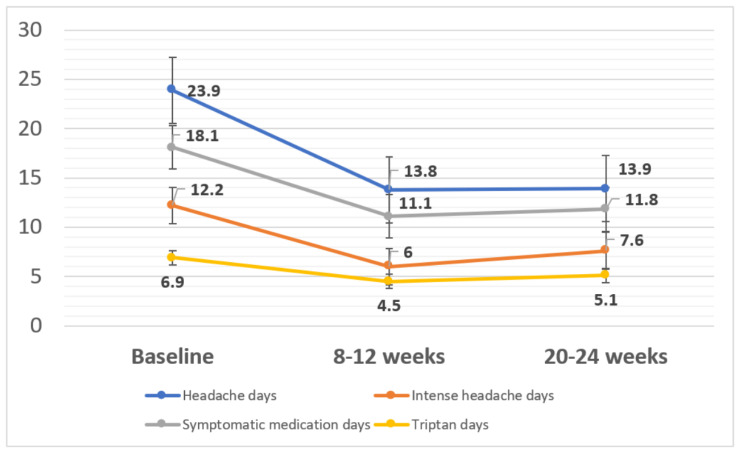
Frequency of headache and acute medication use during the study period. Changes in the headache days and use of symptomatic medication.

**Figure 4 toxins-14-00850-f004:**
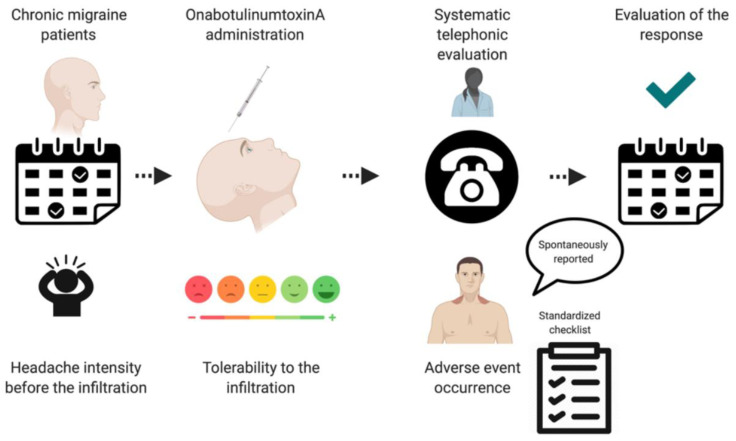
Study protocol.

**Table 1 toxins-14-00850-t001:** Factors potentially associated with a better tolerability. Regression analysis of the possible factors associated with a better tolerability. Numerical Rating Scale (NRS).

*Item*	*Analysis*	*Standardized Beta Coefficient*	*95% CI*	*p-Value*
*Female sex*	Univariate	−0.033	−1.393–0.988	0.737
	Multivariate	−0.063	−1.558–0.796	0.522
*Age (years)*	Univariate	0.222	0.006–0.078	0.023
	Multivariate	0.196	−0.002–0.075	0.061
*Months of CM*	Univariate	0.153	−0.002–0.016	0.118
	Multivariate	0.414	−0.008–0.012	0.680
*Intensity of headache (0–10 NRS)*	Univariate	−0.083	−0.196–0.079	0.400
	Multivariate	−0.135	−0.237–0.046	0.182
*Frequency of headache (days)*	Univariate	0.166	−0.010–0.137	0.090
	Multivariate	0.138	−0.027–0.132	0.196

**Table 2 toxins-14-00850-t002:** Baseline characteristics of patients with the presence or not of adverse events. Comparison between demographical and clinical variables between patients who experienced AEs and those who did not. Chronic migraine (CM).

*Item*	*Adverse Events (n = 75)*	*No Adverse Events (n = 30)*	*p-Value*
*Mean age*	43.6	44.6	0.67
*Months of CM*	40.3	23.8	0.023
*Mean intensity*	3.5	3.3	0.62
*Mean tolerability*	7.6	7.5	0.92
*Headache days per month at baseline*	24.7	22.1	0.026
*Intense headache days per month at baseline*	12.8	10.6	0.11
*Acute medication days per month at baseline*	18.3	17.7	0.75
*Triptan days per month at baseline*	7.3	5.9	0.34

## Data Availability

The database is available for other researchers upon reasonable request.

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
