# Peer review of "Real-World Evaluation of the Tolerability to Onabotulinum Toxin A: The RETO Study"

_toxins, 2022, doi:10.3390/toxins14120850_

Round 1
Reviewer 1 Report
The authors aim to study the tolerability and adverse events that occur following two administrations of Onabot A. The study is simple in design and straightforward in execution.
I only have one significant complaint. Since one of the main goals was to study AEs, the grade of the AEs needs to be reported. There are standard grading scales for each AE. But we are only told that there were no serious AEs, but "serious" was not defined, does that mean grade 3 & 4, or just 4?
Minor editorial points:
Line 30 replace “as long as the” with “because their”
Lines 42-43: “One patient was excluded because it received…” Patients are not “its”, use “they” as a non-gendered pronoun.
Figure 2, what does the * refer to.
Table 3 and figure 3 are totally redundant. One needs to go.
Author Response
Authors’ Response to Reviewers’ Comments
Real-world Evaluation of the Tolerability to Onabotulinum toxin A: The RETO study
We thank the Reviewers for their feedback on our manuscript. We highly appreciate reviewers’ time and their constructive comments. We have thoroughly revised the manuscript following the recommendations made by the Reviewers. All co-authors strongly believe that it has been possible to address the points of critique.
We believe that the changes to the manuscript in this new version have significantly improved the readability and overall quality of the paper.
In this letter the reviewer’s comments are written in bold, while our replies are written in regular font
The changes in the text has been included using Track Changes, following the Journal recommendation.
REVIEWER 1:
The authors aim to study the tolerability and adverse events that occur following two administrations of Onabot A. The study is simple in design and straightforward in execution.
We appreciate your comment
I only have one significant complaint. Since one of the main goals was to study AEs, the grade of the AEs needs to be reported. There are standard grading scales for each AE. But we are only told that there were no serious AEs, but "serious" was not defined, does that mean grade 3 & 4, or just 4?
We appreciate your advice. No 3 or 4 AEs. We have completed in the manuscript (Line 66)
“All AEs were graded 1 (mild but not bothersome) or 2 (bothersome but not dangerous)”
Minor editorial points:
Line 30 replace “as long as the” with “because their”
We agree with your comment; done as suggested
Lines 42-43: “One patient was excluded because it received…” Patients are not “its”, use “they” as a non-gendered pronoun.
Thanks for the correction; done as suggested
Figure 2, what does the * refer to.
Thanks for the remark. It means no-migrainous headache. We have completed as suggested
Table 3 and figure 3 are totally redundant. One needs to go
Thanks again for your comment. You are right. We have removed Table 3

Reviewer 2 Report
This is a well-structured study and a well-written manuscript having great interest.
Here are my few comments:
Please better specify in the introduction and discussion the physiological rationale for Onabotulinum toxin A used for migraine patients and discuss it in the discussion.
Please better specify the definition of chronic migraine in methods.
Please add at the end of the discussion the future prospective deriving from this study.
Author Response
Authors’ Response to Reviewers’ Comments
Real-world Evaluation of the Tolerability to Onabotulinum toxin A: The RETO study
We thank the Reviewers for their feedback on our manuscript. We highly appreciate reviewers’ time and their constructive comments. We have thoroughly revised the manuscript following the recommendations made by the Reviewers. All co-authors strongly believe that it has been possible to address the points of critique.
We believe that the changes to the manuscript in this new version have significantly improved the readability and overall quality of the paper.
In this letter the reviewer’s comments are written in bold, while our replies are written in regular font
The changes in the text has been included using Track Changes, following the Journal recommendation.
REVIEWER 2
This is a well-structured study and a well-written manuscript having great interest.
Thank you very much for your comment
Here are my few comments:
Please better specify in the introduction and discussion the physiological rationale for Onabotulinum toxin A used for migraine patients and discuss it in the discussion.
We appreciate your valuable comment
We have add
Introduction:
OnabotA could decrease peripheral and, indirectly, central sensitization
Discussion:
The exact mechanism by which OnabotA has an antinociceptive effect has not been fully elucidated. Onabot A seems to block peripheral release of neuropeptides involved in neurogenic inflammation as Substance P and CGRP. Besides, it also block the translocation of membrane receptors to the surface of sensory neurons, such as the vanilloid transient receptor potential channel (TRPV1)
And a new reference:
Gago-Veiga AB, Santos-Lasaosa S, Cuadrado ML et al. Evidence and experience with onabotulinumtoxinA in chronic migraine: Recommendations for daily clinical practice. Neurología (Engl Ed) 2019; 34(6): 408-417
Please better specify the definition of chronic migraine in methods.
We appreciate your suggestion. We have completed the definition of CM
(headache occurring on 15 or more days/month for more than three months which, on at least eight days/month has the feature of migraine headache)
Please add at the end of the discussion the future prospective deriving from this study.
We appreciate your advice
We have added a sentence as suggested
In our opinion it is advisable to carry out multicentric studies, with larger samples, and with an evaluation of efficacy depending on whether the procedure was carried out in a patient with or without headache

Reviewer 3 Report
Dear Authors,
In this manuscript, the authors investigated the tolerability of onabotulinumtoxinA administration and the frequency and type of adverse events. We analyzed whether tolerability was associated with the presence of adverse events or with the clinical response.
They found that patients well tolerated onabotulinumtoxinA administration. Patients experienced side effects frequently, but most were local and mild. The most commonly reported side effects were headache and neck stiffness. The authors did not observe the correlation between tolerability to onabotulinumtoxinA administration and the clinical response or the presence of adverse events.
The topic is timely and may attract much attention.
I have only a few suggestions to improve the quality of the manuscript:
1. Introduction:
It would be worthwhile to provide a little more information about chronic migraine:
- Detailed presentation of the disease
- What is the problem with current therapeutic solutions?
- Why is it so important to be able to provide appropriate treatment to those suffering from the disease?
2. The abstract includes the following information: "We included 104 patients, 87.7% female, aged 43.9+/-10.7 years." while the results include the following: "...for a final sample of 105 patients. Mean age was 43.9+/-10.7 years and 92 (87.6%) patients were female." Please unify the data.
3. References:
I recommend authors use more references to support their claims. Thus, I recommend the authors attempt to deepen the subject of their article, as the bibliography is too concise. Nonetheless, in my opinion, less than 50 articles for a research paper are insufficient. Currently, authors cite only 35 papers, and in my opinion, they should mention more than it. I believe that adding more citations will help to provide better and more accurate background to this study.
4. Typing and spacing errors occur in several places. Please read the text carefully and correct these mistakes.
Question:
1. Did the authors observe a difference between male and female patients in the level of tolerance, the appearance of side effects, etc.?
Author Response
Authors’ Response to Reviewers’ Comments
Real-world Evaluation of the Tolerability to Onabotulinum toxin A: The RETO study
We thank the Reviewers for their feedback on our manuscript. We highly appreciate reviewers’ time and their constructive comments. We have thoroughly revised the manuscript following the recommendations made by the Reviewers. All co-authors strongly believe that it has been possible to address the points of critique.
We believe that the changes to the manuscript in this new version have significantly improved the readability and overall quality of the paper.
In this letter the reviewer’s comments are written in bold, while our replies are written in regular font
The changes in the text has been included using Track Changes, following the Journal recommendation.
REVIEWER 3
In this manuscript, the authors investigated the tolerability of onabotulinumtoxinA administration and the frequency and type of adverse events. We analyzed whether tolerability was associated with the presence of adverse events or with the clinical response.
They found that patients well tolerated onabotulinumtoxinA administration. Patients experienced side effects frequently, but most were local and mild. The most commonly reported side effects were headache and neck stiffness. The authors did not observe the correlation between tolerability to onabotulinumtoxinA administration and the clinical response or the presence of adverse events.
The topic is timely and may attract much attention.
I have only a few suggestions to improve the quality of the manuscript:
- Introduction:
It would be worthwhile to provide a little more information about chronic migraine:
- Detailed presentation of the disease
- What is the problem with current therapeutic solutions?
- Why is it so important to be able to provide appropriate treatment to those suffering from the disease?
We highly appreciate your suggestion. We have added the information requested in the first paragraph of introduction and have added a new reference
Blumenfeld AM, Kaur G, Mahajan A et al. Effectiveness and safety of chronic migraine preventive treatments: A systematic literature review. Pain Ther 2022 Epub Ahead of print
- The abstract includes the following information: "We included 104 patients, 87.7% female, aged 43.9+/-10.7 years." while the results include the following: "...for a final sample of 105 patients. Mean age was 43.9+/-10.7 years and 92 (87.6%) patients were female." Please unify the data.
Thank you very much for your correction. It was a mistake in the abstract. We included 105 patients
- References:
I recommend authors use more references to support their claims. Thus, I recommend the authors attempt to deepen the subject of their article, as the bibliography is too concise. Nonetheless, in my opinion, less than 50 articles for a research paper are insufficient. Currently, authors cite only 35 papers, and in my opinion, they should mention more than it. I believe that adding more citations will help to provide better and more accurate background to this study.
We really appreciate your valuable comment. We have increased the number of references
- Typing and spacing errors occur in several places. Please read the text carefully and correct these mistakes.
Thanks for your correction. We have done as suggested though in our word version seems to be less typing and spaces errors
